# A Benzothiadiazole-Based Self-Assembled Cage for Cadmium Detection

**DOI:** 10.3390/molecules28041841

**Published:** 2023-02-15

**Authors:** Zong-Cheng Wang, Ying-Zi Tan, Hui Yu, Wen-Hu Bao, Lin-Li Tang, Fei Zeng

**Affiliations:** Department of Biology and Chemistry, Hunan University of Science and Engineering, Yongzhou 415199, China

**Keywords:** self-assembly, cage, cadmium detection, fluorescent probe, imine condensation

## Abstract

A turn-on fluorescent probe, cage **1**, was efficiently self-assembled by condensing 4,4′-(benzothiadiazole-4,7-diyl)dibenzaldehyde and TREN in chloroform. The formation of cage **1** was characterized and confirmed by NMR spectroscopy, mass spectrometry, and theoretical calculations. The yield of cage **1** could be controlled by tuning the reaction conditions, such as the precursor concentration. Interestingly, the addition of 10 equiv of Cd^2+^ relative to cage **1** could increase the fluorescence almost seven-fold. ^1^H NMR and fluorescence experiments indicating fluorescence enhancement may be caused by the decomposition of cage **1**. Such a high selectivity toward Cd^2+^ implies that the cage could potentially be employed in cadmium detection.

## 1. Introduction

Molecular cages, as one of the supramolecular architectures, provide a synthetic three-dimensional cavity for binding guests, and offer a window for mimicking biological processes in nature [1,2,3,4,5,6,7,8]. Since the first synthesis of cryptands and cavitands in 1969by Jean-Marie Lehn and co-workers [9], various synthetic molecular cages have been reported and have found a variety of potential applications in gas separation [10,11,12,13,14], catalysis [15,16,17,18,19], molecular sensing, and other fields [20,21,22,23,24,25]. Among them, the synthesis of molecular cages by the self-assembly method has attracted extensive attention in recent years due to its simple synthesis and high yield. In 1988, MacDowell and Nelson [26] reported the condensation of tris(2-aminoethyl)amine (TREN) and aldehydes to a make TREN-based [2 + 3] cage in a yield of about 50%. Subsequently, reversible covalent linkages, such as the condensation of amine and aldehydes, have been proved to be a good choice for the high-yield synthesis of cage compounds. Dynamic bonds are formed in a reversible forming/cleaving manner, allowing the self-assembled products to search for their thermodynamic minimum. Recently, Li and coworkers [27] reported the controllable self-assembly of pills and cages via imine condensation for silver cation detection. Later, their group [28] also reported the in situ detection of silver cations in water by a capsule-shaped cage formed by condensing a trialdehyde precursor and TREN. Khashab’s group [29] realized the separation of a mixture of benzene and cyclohexane by thienothiophene cages which were synthesized by a one-step reaction of thieno[2,3-b]thiophene-2,5-dicarboxaldehyde and TREN in acetonitrile via an imine condensation reaction. Moreover, an azobenzene cage [30] for energy efficient and highly selective *p*-xylene separation was also prepared by condensing (E)-4,4′-(diazene-1,2-diyl)dibenzaldehyde and TREN. Although great efforts have been made in the preparation of functional cages, the construction of new cages with novel properties in high yield remains challenging.

Cadmium is an essential resource on earth, widely used in fertilizers and batteries [31,32,33,34,35]. With the increasing demand for batteries, the pollution of the environment with cadmium ions is becoming increasingly serious. Cadmium is harmful to human health and can cause lung, prostate, breast, or endometrial cancer [36,37]. Therefore, developing new methods to detect cadmium in the environment is urgent. Among the various detection techniques, fluorescent probe detection has been proven to be the best choice, not only for its simplicity and low detection limit, but also for its characteristics of intracellular detection [38,39,40]. Although considerable efforts have been devoted to Cd^2+^ fluorescent probes, the development of selective fluorescent probes for Cd^2+^ remains a great challenge due to it being easily interfered with by other transition metals, especially Zn^2+^ ions in the same group [41,42,43].

Recently, Li and coworkers reported the synthesis of a benzothiadiazole-based macrocycle and found that the intense fluorescence of the macrocycle in the solid state was higher than the monomer [44]. We questioned whether the intense fluorescence of the benzothiadiazole-based cage would increase compared to the monomers. Herein, as a continuation of our interest for supramolecular chemistry [45,46,47,48,49], we report a turn-on fluorescent probe based on a self-assembled cage for the selective detection of Cd^2+^. The self-assembled cage **1** was obtained by the condensation of 4,4′-(benzothiadiazole-4,7-diyl)dibenzaldehyde and TREN in chloroform via the [3 + 2] approach (Figure 1). A low precursor concentration can improve the yield of self-assembled cage **1**. Interestingly, cage **1** showed a selective response to cadmium ions. The fluorescence intensity of cage **1** was increased almost seven-fold after the addition of cadmium ions, implying that the self-assembled cage represents a promising fluorescent probe for cadmium ion detection.

## 2. Results and Discussion

First, 4,4′-(benzothiadiazole-4,7-diyl)dibenzaldehyde **2** was prepared according to the reported literature [50]. TREN was used as received without further purification. As shown in Figure 1II, after mixing **2** (1.55 mg, 7.5 mM) and TREN (0.44 mg, 5 mM) in CDCl_3_ (0.6 mL) at room temperature for 12 h, a small amount of yellow precipitate was formed and the ^1^H NMR spectrum of the mixture showed one set of new signals that were different from those of **2**, suggesting that cage **1** with a relatively high symmetry was formed. However, according to the ^1^H NMR spectrum results (Figure 1I),compound **2** was not completely converted into cage **1**, and a large amount of yellow precipitate would be generated if the reaction time was further prolonged. These results made us doubt whether **2** and TREN could be completely converted to cage **1**. We then studied the self-assembly of **2** and TREN at low concentrations. To our surprise, when **2** (1.24 mg, 6 mM) and TREN (0.36 mg, 4 mM) were combined in CDCl_3_ (0.6 mL) at room temperature for 12 h, only a set of relatively sharp resonances of cage **1** were observed in the ^1^H NMR spectrum, indicating that **2** and TREN were completely converted to cage **1**, and the yield was quantitative (Figure 1III). Protons a-c corresponding to cage **1** shifted remarkably up field (Δ*δ* ≈ 1.82 ppm, 0.61 ppm, 0.71 ppm, and 1.23 ppm, respectively) compared with protons a-c corresponding to **2**, indicating that the benzothiadiazole moiety experienced a shielded magnetic environment in the cavity of cage **1**. The structure of cage **1** was further confirmed by two-dimensional NMR spectroscopy and mass spectrometry (see Appendix A). Moreover, at a lower concentration, **2** (0.62 mg, 3 mM) and TREN (0.18 mg, 2 mM) in CDCl_3_ (0.6 mL) could still completely self-assemble into cage **1** (Figure 1IV).

We attempted to obtain a single crystal of cage **1** that is suitable for X-ray diffraction analysis; however, this was unsuccessful. Density functional theory (DFT) methods were used to gain further insights into the formation of cage **1** by using Gaussian 09 software and choosing6-311G as the basis sets. In the optimized structure (Figure 2), the three benzothiazole units of cage **1** are oriented in the manner of a three-blade propeller which may reduce the repulsive force between the benzothiazole units and facilitate the formation of the cage.

In order to expand the application of cage **1**, we first tested the coordination properties of cage **1** withvarious metal cations by fluorescence experiments. A number of metal cations, such as Co^2+^, Ba^2+^, Pb^2+^, Mg^2+^, Zn^2+^, Fe^2+^, Ni^2+^, Ag^+^, Cu^2+^, and Cd^2+^, were added to a solution of cage **1**. As shown in Figure 3, it was found that the fluorescence of cage **1** is slightly quenched after the addition of Ag^+^ or Cu^2+^, which is different from other results where Ag^+^ can completely quench the fluorescence of the TREN-based cage [27]. In contrast, Co^2+^, Ba^2+^, Pb^2+^, Mg^2+^, Zn^2+^, Fe^2+^, Ni^2+^, or Cd^2+^ are able to enhance the fluorescence of cage **1**. It is noteworthy that the addition of 10 equiv of Cd^2+^ relative to cage **1** could increase the fluorescence by almost seven times, while other cations could only slightly increase the fluorescence of cage **1**. This enhancing behavior might potentially be employed to the selective detect of cadmium ions without interference by zinc and other cations. Moreover, ^1^H NMR experiments were further carried out to investigate the mechanism of Cd^2+^ to enhance the fluorescence of the cage. As shown in Appendix A, after the addition of 0.5 equiv of Cd^2+^ cation, the protons corresponding to cage **1** shifted remarkably downfield and a small amount of TREN was formed, suggesting that decomposition of the cage may have occurred. Moreover, the fluorescence of cage **1** that is coordinated with the Cd^2+^ cation is similar to that of compound **2** (Figure 3d), indicating that the fluorescence enhancement may be caused by the decomposition of cage **1**. To the best of our knowledge, this system is the first example of cadmium detection based on self-assembled cage formed by tris(2-aminoethyl)amine (TREN) and aldehyde condensation.

## 3. Materials and Methods

### 3.1. General Considerations

Unless otherwise noted, all reagents were obtained from commercial suppliers and used without further purification. ^1^H NMR, ^13^C NMR spectra were recorded on a Bruker DMX400 NMR spectrometer. 4,4′-(Benzothiadiazole-4,7-diyl)dibenzaldehyde **2** was prepared according to the reported literature [50]. Electrospray ionization mass spectra (ESI-MS) were recorded on a Thermo Fisher^®^Exactive LC-MS spectrometer (Thermo Fisher Scientific, Waltham, MA, USA).

### 3.2. Typical Procedure for the Synthesis of Cage 1

Cage **1** was obtained by condensing **2** (1.24 mg, 6 mM) and tris(2-aminoethyl)amine(TREN) (0.36 mg, 4 mM) in CDCl_3_ (0.6 mL). The solution was sealed in an NMR tube for 12 h without stirring to allow the system to reach equilibrium. ^1^H NMR (400 MHz, Chloroform-*d*) δ 8.34 (s, 6H), 7.59 (d, *J* = 8.0 Hz, 12H), 7.39 (d, *J* = 8.0 Hz, 12H), 6.71 (s, 6H), 3.91 (s, 12H), 2.90 (s, 12H). ^13^C NMR (101 MHz, CDCl_3_) δ 161.96, 153.58, 137.89, 136.15, 132.17, 129.05, 128.41, 127.11, 57.90, 52.22. ESI-MS cald. for [M + Na]^+^: 1239.42, found: 1239.15.

## 4. Conclusions

In summary, a turn-on fluorescent probe, cage **1**, was self-assembled by condensing 4,4′-(benzothiadiazole-4,7-diyl)dibenzaldehyde and TREN via imine condensation in chloroform. The yield of cage **1** could be improved by lowering the precursor concentration. Moreover, the addition of cadmium ions could remarkably enhance the fluorescence intensity of the system by decomposing cage **1**. An easy preparation of cage **1** and a selective turn-on fluorescent sensor for Cd^2+^ over other cations were achieved using the present system, which is promising for practical applications in the selective detection of Cd^2+^ in petrochemical industry wastes. Further work will focus on the development of a water-soluble self-assembled cage for the detection of Cd^2+^ in water and living cells.

## Data Availability

Not applicable.

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
