# Peer review of "A Benzothiadiazole-Based Self-Assembled Cage for Cadmium Detection"

_molecules, 2023, doi:10.3390/molecules28041841_

Round 1

Reviewer 1 Report

The manuscript entitled "Benzothiadiazole-based self assembled cage for cadmium detection" describes that a [2+3] cage via imine condensation can be used as a fluorescence probe for the detection of Cd2+The authors demonstrated that the cage undergoes fluorescence quenching upon addition of either Ag+ or Cu2+. In contrast, upon addition of other cations, the luminescence of the cage increased, among which Cd2+ led to the most remarkable fluorescence TURN-ON. The paper could be published, after the authors addressed all the following questions.

1.    The authors discovered that at higher concentration, the conversion from aldehyde material to cage is not complete, and they found unreacted aldehyde precursor, while at lower concentrations, the formation of cage is close to quantitative. However, in higher concentration, the equilibrium should shift to the side of cage in the context of thermodynamics. And in the context of dynamics, raising the concentration will increase the cage formation rate. The implication is that, the argument might be incorrect here. It is highly possible that the reactant stoichiometry is not accurate, i.e., the TREN/aldehyde ratio is lower than 2:3. The authors should repeat the experiment and shed more light on this issue.

2.    The mechanism of Cd2+ to enhance the fluorescence of the cage should be unraveled. The authors should provide the 1H NMR spectra to shed more light on it. The fluorescence spectrum of the aldehyde precursor should also be introduced. If the starting material has stronger fluorescence relative to the cage, it is also likely that Cd2+ decomposed the cage.

Author Response

  1. According to the suggestion, we repeated the preparation of cage 1 by the mixture of 2 (1.55 mg, 7.5 mM) and TREN (0.44 mg, 5 mM) in CDCl3 (0.6 mL) at room temperature for 12h and a small amount of yellow precipitation was formed. The 1H NMR spectrum of the mixture showed that there was still unreacted aldehyde precursor. However, after reduce the reaction time to 8 hours, 2 and TREN can be completely transformed into cage 1. Thus, “a small amount of yellow precipitation was formed.”was added before the “the 1H NMR spectrum of the mixture showed one set of new signals different from those of the 2, suggesting that a relatively high symmetry cage 1 was formed. ”
  2. According to the suggestion, the 1H NMR spectra of cage 1 with Cd2+ were added in Fig. S7. The fluorescence spectrum of the aldehyde precursor was added in Figure 3. As shown in Fig. S7, after addition of 5 equiv of Cd2+ cation, the protons b corresponding to cage 1 shifted remarkably downfield and a small amount of TREN was formed, suggesting thedecomposed of cage may have occurred. The fluorescence of cage 1 that coordination with Cd2+ cation is similar to that of compound 2, so the fluorescence enhancement may be caused by the decomposed of cage 1.

Reviewer 2 Report

The authors synthesized [3+2]-type cage molecule 1 by condensing 4,4'-(benzothiadiazole-4,7-diyl)dibenzaldehyde and tris(2-aminoethyl)amine (TREN), and investigated the metal ion responsiveness of PL behavior. Selective capture of metal ions is one of the important issues, and detection of cadmium ions with fluorescent probes is particularly important because of its simplicity and low detection limit. Therefore, the content described in this manuscript is one of the subjects of interest for readers of Molecules, and this reviewer believes that it will be of interest to more readers. Therefore, this manuscript is suitable for publication in Molecules. However, this reviewer would like to request comments or answers on the following points, and after confirming the comments and answers, this reviewer would like to provide acceptance for publication.

(1) In general, the condensation reaction of aldehydes and primary amines is an equilibrium reaction, releasing H2O out of the system and tilting the equilibrium towards the imine production system. This reviewer thinks it is common to carry out the condensation reaction using a dehydrating agent. However, in this experiment, the reaction was carried out in a CDCl3 solvent in an NMR tube. As long as H2O is not released, this reviewer does not think the equilibrium will be biased toward the imine production system no matter how long the mixture is mixed. Comments are requested on this point.

(2) In this study, the target molecule is the cage molecule of benzothiadiazole and TREN. The reason for using TREN is understandable, but the reason for using benzothiadiazole is unclear. Explain the rationale for choosing benzothiadiazole, that is, why the authors chose benzothiadiazole.

(3) Figure 3a: The PL spectrum of the “cage 1” molecule is obscured by other spectra. It is understandable to want to appeal the trapping of Cd2+, but it is unkind to discuss the trapping effect of metal ions without showing the unique photophysical properties of cage 1 (absorption spectrum and PL spectrum). Absorption and PL spectra of Cage 1 should be clarified in the text, after which metal trapping effects should be discussed.

(4) Depending on the presence or absence of metal ion, the maximum PL wavelength of the PL spectrum changes, shifts to longer wavelengths, and shifts to shorter wavelengths. Further consideration should be given to trapping metal ions and PL spectral changes from the viewpoint of structural change of cage molecule. In addition, no discussion is given to the fact that when Cd2+ is captured, the emission maximum wavelength is shifted and the emission intensity is remarkably improved. Since this manuscript only describes the results, this section is "Results", not "Results and Discussion", so more detailed discussion needs to be added. It is reported as a "Communications", but the content is important, but it is not very prompt, so it should be reported as an "Research Article" after adding detailed considerations.

Author Response

1.      The dynamic bonds form in a reversible forming/cleaving manner, allowing the self assembled products to search for their thermodynamic minimum. The implication is that a target molecule could be produced in high yields in some cases, even to a quantitative yield when the compound represents the most thermodynamically favored product. Some reported imines-based cages also prepared in CDCl3 solvent or water in an NMR tube with high yields. (see Org. Lett. 2018, 20, 7447-7450. Inorg. Chem. 2022, DOI: 10.1021/acs.inorgchem.1c03825)2.      2,1,3-Benzothiadiazole (BT) based molecules are ideal chromophores and have been widely used as building block in luminescent materials. Therefore, we use 4, 4’-(benzothiadiazole-4,7-diyl)dibenzaldehyde as building block to construct luminescent cage 1.3.      According to the suggestion, Absorption and PL spectra of Cage 1 have been added in Figure 3f and 3g. 4.      According to the suggestion, the mechanism of Cd2+ to enhance the fluorescence of the cage was investigated by 1H NMR experiments and the discussion was added in the manuscript. After adding Cd2+ ions, the cage decomposes to produce compound 2, resulting in fluorescence enhancement and the maximum PL wavelength of the PL spectrum shifts to shorter wavelengths. 

Round 2

Reviewer 2 Report

Some answers to this reviewer’s comments were satisfactory, but some answers remain questionable.

It is quite understandable that benzothiadiazole is used as a great luminophore. However, the answer "Therefore, we use 4, 4’-(benzothiadiazole-4,7-diyl)dibenzaldehyde as building block to construct luminescent cage 1." is too naive. In the opinion of this reviewer, in order to publish in Molecules, which has a high impact factor, appropriate molecular design and the correlation between the molecular structure and the obtained properties should be extensively discussed. In this regard, this reviewer strongly requests further improvement, such as adding appropriate molecular design to the text.

After adding Cd2+ ions, the cage decomposes to produce compound 2, resulting in fluorescence enhancement and the maximum PL wavelength of the PL spectrum shifts to shorter wavelengths”… In order to discuss the PL enhancement by the addition of Cd2+, the authors conducted 1H NMR experiments and concluded that the addition of Cd2+ caused the decomposition of cage 1 and the regeneration of starting 2. What do the authors think about the following points:

(1) Is it correct that "cage molecule 1 does not capture Cd2+"?

(2) In other words, is it correct to understand that in cases other than Cd2+, the PL intensity does not increase because the cage molecule 1 does not decompose?

(3) If the above understanding is correct, is it correct to understand that the results of Job's Plot described in the manuscript before revision indicate that the cage molecule 1 decomposes and forms a 1:1 complex of TREN and Cd2+?

(4) If the above understanding is correct, please add the following points.

・The reviewer does not understand the need to use cage molecules for this Cd2+ detection. Please explain the implications of using cage molecules for this Cd2+ detection.

・Please describe the proposed mechanism why Cd2+ specifically decomposes the cage molecule. Conversely, please explain why it does not decompose with metal ions other than Cd2+.

Minor point:

Lines 111–114 in the revised manuscript: “As shown in Fig. S7, …, the protons b corresponding to cage 1 …”: Isn't this proton a, not proton b?

Author Response

Reviewer 2:

  1. According to the suggestion, we added the reason for choosing 4, 4’-(benzothiadiazole-4,7-diyl)dibenzaldehyde as building block to construct the cage. Recently, Li and coworkers reported the synthesis of benzothiadiazole-based macrocycle and found that the intense fluorescence of macrocycle in the solid state is higher than monomer (ref. 44 Commun. 2022, 13, 2850). We wonder that whether the intense fluorescence of benzothiadiazole-based cage will increase in comparison with the monomer. Therefore, we choose 4, 4’-(benzothiadiazole-4,7-diyl)dibenzaldehyde as building block for constructing the cage.

2.      According to the results of 1H NMR experiments, cage molecule 1 does not capture Cd2+. Moreover, the fluorescence of cage 1 that after addition of Cd2+ is similar to that of compound 2, indicating the fluorescence enhancement may be caused by the decomposed of cage 1.

3.      It is correct to understand that the PL intensity of cage 1 does not increase after addition of other cations because the cage molecule 1 does not decompose.

4.      It is correct to understand that the results of Job's Plot indicate that the cage molecule 1 decomposes and forms a 1:1 complex of TREN and Cd2+.

5.      Cage molecules provide synthetic cavities to effectively bind guest molecules by multiple noncovalent interactions with the cage-building blocks surrounding the cavity. The inclusion of guest molecules in suitable host-cage derivatives results in restricted systems that confer entrapped substrates novel properties. According to the cavity size of the cage 1 and the coordination sites contained in it, we hope to use it to detect metal ions. It is found that it has good specific recognition effect for cadmium ion.

6.      The radius of cadmium ion matches the cavity of the cage 1, and cadmium ion can enter the cavity of the cage. However, after entering, the cage is unstable and the cage is decomposed. Other metal ions, such as Ba2+ and Pb2+ ions have a large radius and cannot enter the cavity of the cage, so the cage will not be decomposed. Ag+ ions can enter the cavity of the cage to form complex 1-Ag+ and quench the fluorescence of cage 1.

7.      According to the suggestion, “the protons b” has been changed to “the protons a”.